# The Effect of Vitamin D3 and Silver Nanoparticles on HaCaT Cell Viability

**DOI:** 10.3390/ijms23031410

**Published:** 2022-01-26

**Authors:** Samuela Cataldi, Maria Rachele Ceccarini, Federica Patria, Tommaso Beccari, Martina Mandarano, Ivana Ferri, Andrea Lazzarini, Francesco Curcio, Elisabetta Albi

**Affiliations:** 1Department of Pharmaceutical Sciences, University of Perugia, 06126 Perugia, Italy; samuela.cataldi@unipg.it (S.C.); mariarachele.ceccarini@unipg.it (M.R.C.); patriafederica@gmail.com (F.P.); tommaso.beccari@unipg.it (T.B.); 2Division of Pathological Anatomy and Histology, Department of Experimental Medicine, School of Medicine and Surgery, University of Perugia, 06126 Perugia, Italy; martina.mandarano@unipg.it (M.M.); ivana.ferri@unipg.it (I.F.); 3Crabion S.R.L., 06100 Perugia, Italy; andrylazza@gmail.com; 4Institute of Clinical Pathology, Department of Medicine, University of Udine, 33100 Udine, Italy; francesco.curcio@uniud.it

**Keywords:** wound healing, cholecalciferol, silver nanoparticles, sphingomyelinase, keratinocytes

## Abstract

Vitamin D3, known to regulate bone homeostasis, has recently been shown to have many pleiotropic effects in different tissues and organs due to the presence of its receptor in a wide range of cells. Our previous study demonstrated that vitamin D3 was able to increase the wound healing respect to the control sample, 24 h after cutting, without however leading to a complete repair. The aim of the study was to combine vitamin D3 with silver nanoparticles to possibly enable a faster reparative effect. The results showed that this association was capable of inducing a complete wound healing only after 18 h. Moreover, a treatment of vitamin D3 + silver nanoparticles yielded a small percentage of keratinocytes vimentin-positive, suggesting the possibility that the treatment was responsible for epithelial to mesenchymal transition of the cells, facilitating wound healing repair. Since vitamin D3 acts via sphingolipid metabolism, we studied the expression of gene encoding for the metabolic enzymes and protein level. We found an increase in neutral sphingomyelinase without involvement of neutral ceramidase or sphingosine kinase2. In support, an increase in ceramide level was identified by Ultrafast Liquid Chromatography–Tandem Mass Spectrometry, suggesting a possible involvement of ceramides in wound healing process.

## 1. Introduction

The production of vitamin D3 or cholecalciferol occurs in the skin by ultraviolet radiation from sunlight starting from its precursor 7-dehydrocholesterol, located in the keratinocyte membranes of the basal and spinous layer of epidermis [1]. The active hormonal form or calcitriol (1,25-dihydroxyvitamin D) is synthesized by hydroxylation in the liver and kidney [1]. Interestingly, in keratinocyte, vitamin D3 can be directly metabolized to 25-hydroxyvitamin D3 and further metabolized to calcitriol [2]. Thus, keratinocytes are the only cells in the body containing the entire pathway of vitamin D3 and here calcitriol negatively regulates its own levels [3]. This is relevant considering that keratinocytes are located in the stratum basale of epidermis that rests on the basal lamina separating the dermis and epidermis. Here the cells proliferate and then differentiate as they migrate to the upper, acquiring the characteristics of a fully differentiated corneocyte, which eventually dies of apoptosis. Thus, keratinocytes have proliferative, differentiative, and apoptotic capacities, and many studies on the effect of vitamin D3 in keratinocytes have led to different results. In the 1990, Matsumoto et al., showed an anti-proliferative effect of calcitriol on keratinocytes [4]. Subsequent studies demonstrated that physiological doses of calcitriol were able to regulate keratinocytes differentiation [5] and pharmacological concentrations exerted a pro-apoptotic effect in keratinocytes [6]. Moreover, calcitriol was shown to have potent anti-inflammatory and anti-cancer increasing in the skin [7]. We have previously shown that calcitriol was capable of promoting wound repair without leading to the entire closure of the cut [8]. It is known that wound repair takes place thanks to the proliferation of keratinocytes and their transdifferentiation to mesenchymal cells epithelial–mesenchymal transition (EMT) [9]. Among EMT forms, the type 2 is specific of wound healing [10].

Therefore, finding a substance that could help vitamin D3 stimulate keratinocyte proliferation and transdifferentiation for the complete wound repair remained an open challenge. It has been demonstrated that silver nanoparticles (AgNPs) are useful for different medical applications as medical products, medical instruments, water filtration, personal hygiene products, and others [11]. AgNPs are well known for their unique physical, chemical, and biological properties, which depend on their size, shape, and surface charge [12]. Thanks to their size (1–100 nm), the particles enter cells mostly through endocytosis and release of ions into the cytoplasm [13]. The antibacterial, antiviral, antifungal, antiprotozoal, antiparasitic, and anticancer efficacy of AgNPs has been demonstrated [12]. Interestingly, AgNPs also have antioxidant properties [14]. Considering that the antimicrobial agents that facilitate the repair processes [15], the idea that vitamin D3 could be helped by AgNPs in repairing the wound was considered. On the other hand, although AgNPs are widely used in basic and applied scientific research, their combination with vitamin D3 has never been studied.

However, we wanted to study the reparative effect of a wound in a system that did not include microbial agents to analyze a new functional aspect of the AgNPs. Therefore, the study was aimed to investigate whether AgNPs could help vitamin D3 in wound repair and what the molecular mechanisms involved in the response to vitamin D3-AgNPs treatment of keratinocytes were.

## 2. Results

### 2.1. Action of VD3 and AgNPs on HaCaT Cell Viability

In initial studies, we set out to determine the effects of VD3 or AgNPs and their combination on HaCaT cell viability. It was reported that 100 nM VD3 was a physiological dose [16] and that AgNPs at 25 ppm, 2.5 ppm, and 0.25 ppm do not have a toxic effect in a model of human reconstructed epidermis [17]. Therefore, we have chosen to use 100 nM VD3 and 5 ppm AgNPs as minimum doses. Increased doses were selected to test their effect on cell viability (Figure 1a). As can be seen, 100 nM, 133 nM, and 166 nM VD3 induced more than 80% cell viability; 1% and 2% DMSO were used as positive controls. Since HaCaT is an immortalized cell line, the slight reduction of cell viability upon VD3 treatment in comparison to the control sample is acceptable and the standard deviation conforms to the experimental system [18]. Five ppm and 6.25 ppm AgNPs did not induce significant variations respect to control sample, and with 7.5 ppm AgNPs the cells were more vital than control. Trypan blue exclusion assay showed no significant changes of live cell numbers and dead cell numbers at all concentrations, with the exclusion of 100 nM VD3 + 6.25 ppm AgNPs, 166 nM VD3 + 6.25 AgNaPs, and 166 nM VD3 + 7.5 ppm AgNPs treatments that induced a reduction in live cells and an increase in dead cells with respect to the control cells (Figure 1b). The 133 nM VD3 + 5 mpp AgNPs combination induced the best effect in terms of cell viability, live cell and dead cell numbers. Therefore, we decided to use this combination for all subsequent experiments.

### 2.2. Effect of Vitamin D3 and Silver Nanoparticles on Wound Healing: Molecular Mechanism Involved

Since we observed a good effect of 133 nM VD3 + 5 mpp AgNPs combination on keratinocyte viability, it became interesting to test its effect on keratinocyte proliferation in a simulated in vitro repair following the reproduction of a wound. Thus, the specific system of wound healing was assessed. An open gap of 0.9 mm was created by a “wound”, as reported in the Section 4. The reaction of the cells to fill the damaged area was inspected microscopically over time (6, 12, and 18 h) (Figure 2). In the control sample, the open gap remained unchanged after 6 h (0.9 mm), with a reduction of 25% and 60% after 12 h and 18 h, respectively. Upon 133 nM VD3 treatment, the open gap reduced 36% after 6 h, remained unchanged after 12 h, and after 18 h induced a repair of the 77%. Upon 5 ppm AgNP treatment, the wound repair was very similar to that obtained with VD3. Interestingly, the combination of 133 nM VD3 and 5 ppm AgNP stimulated complete wound closure only after 18 h. Cellular hypercolourability as evident in the closing line.

To investigate the possible molecular mechanisms involved in VD3 + AgNPs wound repair, we analyzed the expression of cytokeratin as an epithelial marker and vimentin as a mesenchymal marker [10] in cells after 24 h from treatment (Figure 3). In line with this, our results showed a high amount of the cells positive to cytokeratin and negative to vimentin in control and VD3 + AgNPs treated samples. However, interestingly, we also observed a low cell number negative to cytokeratin and a low cell number positive to vimentin only in VD3 + AgNPs treated samples, indicating a differentiation of some keratinocytes with their epithelial-mesenchymal transition useful for the wound healing [9,10]. As a negative control of immunohistochemical analysis, we used α-smooth muscle actin (α-SMA, Figure 3).

It is known that vitamin D3 stimulates its own receptor (VDR) [19]. Thus, we analyzed VDR gene expression and protein level. The results showed that VD3 combined with AgNPs did not affect VDR ene expression (Figure 4a). Differently, the treatment induced an increase in VDR protein level (Figure 4b,c), indicating a possible slowdown of VDR breakdown due to its intracellular need to promote the response to vitamin D3.

To further investigate the molecular mechanisms that take place during wound healing in response to the VD3-AgNPs treatment, we studied the possible involvement of sphingomyelin (SM) pathway due to its central regulatory role in the response to VD3 for cell signaling [20,21]. Our previous research showed the increase in neutral sphingomyelinase (nSMase) gene and protein expression during the incomplete wound healing, VD3 induced [8].

Therefore, we decided to investigate nSMase and nCerase as molecules involved in keratinocyte proliferation [8,22]. To this end, gene and protein analyses were performed. Basally, SMPD4 gene encoding for nSMase [8] and ASAH2 gene encoding for nCerase was expressed in keratinocytes (Figure 4a). Interestingly, and unexpectedly, VD3 + AgNPs treatment overexpressed only SMPD4. These latter results prompted us to evaluate the protein expression. The results showed an upregulation of nSMase and a low reduction of nCerase (Figure 4b,c) by suggesting an increase of Cer level. Then, thought we would investigate the behavior of SPHK2 that increases the Cer level thanks to the savage pathway [23]. Levels of SPHK2 gene expression remained mostly unchanged when compared to control (Figure 4a). Differently, SPHK2 protein was slightly reduced (Figure 4b,c).

To confirm the hypothesis of Cer accumulation upon VD3 + AgNPs treatment, we performed a lipidomic analysis by UFLC-MS/MS. We analyzed Sph species by using 12:0 SM, 16:0 SM, 18:1 SM, 24:0 SM, 6:0 Cer, 8:0 Cer, 16:0 Cer, 18:0 Cer, 20:0 Cer, and 24:0 Cer, as external calibrators. The results showed that 12:0 SM was undetectable in both control and experimental samples. Therefore, we added to the samples 12:0 SM as internal standards. The recovery of 12:0 SM was 87% and 84% in the control and experimental cells, respectively. Similarly, 6:0 Cer and 8:0 Cer were undetectable in both control and experimental samples; 18:0 Cer was present in traces. Thus, we added 6:0 Cer to the samples as internal standards. The recovery of 6:0 Cer was 85% and 86% in the control and experimental cells. Figure 5 shows a significant decrease in 16:0 SM and 24:0 SM. Consistent with loss of SM, levels of 16:0 Cer, 20:0 Cer, and 24:0 Cer were significantly increased.

## 3. Discussion

VD3 has been implicated in bone metabolism, but recently it has been identified as a potential contributor to the extraosseous pathophysiological conditions in many tissues and organs, including brain [17,24,25], kidney [26], immune system [27], skin [28], and others. Our previous results showed accelerated wound repair with VD3 but without complete repair [8]. As a consequence, we thought of combining VD3 treatment with AgNPs, which is known to have an effect in skin repair [29,30].

Therefore, we first looked for the optimal combination dose of VD3 with AgNPs in terms of cell viability and mortality and then we used the identified dose to study its effect on wound repair. Here, we demonstrated for the first time the need of VD3 coupled AgNPs for an effective wound healing. A complete repair was achieved quickly, only after 18 h from treatment. Additionally, we showed, by vimentin expression, that VD3-AgNPs was able to drive the keratinocyte differentiation into mesenchymal cells, thereby promoting the structure of wound repair. As VD3 alone accelerated wound repair with the degradation of SM, without leading to complete closure [8], we investigated the SM pathway upon VD3-AgNPs treatment that induced the complete closure. Interestingly, we showed a strong upregulation of nSMase due to the overexpression of the gene encoding for this enzyme. As a consequence, we supposed the SM hydrolysis with Cer generation that could be catabolized by ceramidase (Cerase), to form free fatty acids and sphingosine, that in turn could be phosphorylated by sphingosine kinase (SPHK) to form sphingosine 1 phosphate, a known cellular mediator [31]. Thus, in exploring this pathway we focused the attention on Cerase and SPHK.

The family of Cerases includes acid Cerase, neutral Cerase (nCerase), and three isoforms of alkaline Cerase (1,2,3) encoded by five different genes such as *ASAH1*, *ASAH2*, *ACER1*, *ACER2*, and *ACER3*, respectively [32]. aCerase and nCerase are ubiquitous. aCerase localizes to the lysosome and nCerase ASAH2 is a membrane protein. Among alkaline Cerase, the isoform 1 is a protein of the endoplasmic reticulum, the isoform 2 is associated with Golgi apparatus, and isoform 3 is localized in both endoplasmic reticulum and Golgi apparatus [33]. It has been demonstrated that the upregulation of alkaline Cerase 1 induced growth arrest and the differentiation of human epidermal keratinocytes playing a fundamental role in skin barrier formation [34]. Houben et al., reported that keratinocytes express all five Cerase isoforms; aCerase and alkaline Cerases increased significantly with cell differentiation. Differently, nCerase predominated in proliferating keratinocytes [22]. Our data showed that only few keratinocytes differentiated upon VD3-AgNPs treatment, while many cells rapidly proliferated leading to a wound repair in very short time. Therefore, we studied the behavior of nCerase that resulted in a reduced level without changes of *ASAH2* gene expression, suggesting a possible, accelerated breakdown due to the need to leave a high level of Cer.

The family of SPHKs includes SPHK1 and SPHK2. The first is localized in cell membranes and the second in the nucleus and endoplasmic reticulum [35]. Interestingly, SPHK1 decreases the level of Cer and SPHK2 increases it [35]. In fact, SPHK1 inhibits Cer biosynthesis and SPHK2 plays a role in a sphingosine salvage pathway, by increasing Cer level [34]. Thus, we observed that the change of SPHK2 showed a response to VD3-AgNPs treatment similar to that of Cerase. We speculated that the slight reduction in the protein level of SPHK2, without changes in gene expression, could be due to an increase in its breakdown for the non-necessity of the salvage pathway. Importantly, the sphingolipidomic analysis had confirmed the catabolism of SM with production of Cer. On the other hand, the high levels of Cer testify to its lack of use by other enzymes.

It is possible that the high level of nSMase, with a reduction in SM and an increase in Cer content, might be the signal transduction pathway used by VD3 + AgNPs to promote keratinocyte proliferation, supporting previous studies showing the relation nSMase-cell proliferation [36]. The current results added a new dimension to the regulation of keratinocyte proliferation via Cer, considering that Cer is essential for skin health [37,38].

### Conclusions

Taken together, these results suggest that VD3 + AgNPs combination is essential for a complete wound repair in a short time and that it involves the SM pathway as the signal transduction. These findings further extend our current knowledge of VD3 and AgNPs in biological and cellular events. From a therapeutic point of view, our results could have important implications for the skin repair after damage.

## 4. Materials and Methods

### 4.1. Experimental Plan

The study’s design was reported in the Figure 6.

### 4.2. Reagents

Dulbecco’s modified Eagle’s medium (DMEM), L-glutamine, trypsin, and ethylenedi-aminetetraacetic acid disodium and tetra-sodium salt (EDTA) were from MicrotechSrl (Pozzuoli, NA, Italy). Fetal bovine serum (FBS), penicillin–streptomycin, and 6× loading dye were obtained from Thermo Fisher Scientific (Waltham, MA, USA). Antibiotics, Dulbecco’s phosphate buffer saline (PBS) pH 7.4, and agarose were from Invitrogen Srl (Milan, Italy). Dimethylsulfoxide (DMSO), ethanol, hydrochloric acid, sodium chloride, and sodium hydroxide were purchased from Carlo Erba Reagenti srl (Milan, Italy). Trypan blue solution 0.4%, ethidiumbromide, formaldehyde, solution tris(hydroxymethyl)aminimethane (Tris), 3-[4,5-dimethyl-2-thiazolyl]-2,5-diphenyl-2-tetrazoliumbromide (MTT), and 1α,25(OH)2VD3 were obtained from Sigma-Aldrich (now Merck, Darmstadt, Germany). Fixing solution and cell stain solution were from Cell Biolabs, INC (San Diego, CA, USA). RNAqueous^®^-4PCR kit was from Ambion Inc. (Austin, TX, USA). anti-VDR was from Elabscience (Houston, TX, USA). Anti-nSMase, anti-nCerase, anti-SPK2anti-βtubulin, and goat anti-rabbit secondary antibodies were from Abcam (Cambridge, UK). TaqMan-Array 96-well plates were purchased from Applied Biosystems (Foster City, CA, USA). RNAqueous^®^-4PCR kit was from Ambion Inc. (Austin, TX, USA). SDS-PAGE molecular weight standards were purchased from Nzythech (Lisboa, Portugal). Chemiluminescence kits were purchased from Amersham (Rainham, Essex, UK). Silver nanoparticles (AgNPs) with an average size of approximately 30 nm (more than 80%) were provided by NanoBMat Company (Düsseldorf, Germany).

### 4.3. Cell Culture and Treatments

Human keratinocytes (HaCaT cell lines) were purchased from I.Z.S.L.E.R. from the Istituto Zooprofilattico Sperimentale della Lombardia e dell’Emilia Romagna ‘Bruno Ubertini’ (Brescia, Italy). Cells were grown in 75 cm^2^ tissue flasks with DMEM medium supplemented with 100 U/mL penicillin, 100 μg/mL streptomycin, 2 mM L-glutamine, and 10% FBS under a humidified atmosphere of 5% CO_2_ at 37 °C. After colonies were formed (80–90% of confluence), plates were washed with PBS 1× and harvested by 0.05% trypsin in 0.02% Na_4_EDTA for 10 min at 37 °C. Cells were treated with different doses of VD3 dissolved in absolute ethanol as vehicle (100 nM, 133 nM, and 166 nM) or highly stable aqueous colloidal solution of AgNPs of 30–50 nm of diameter. The starting AgNPs concentration was 1 mg/mL, corresponding to 1000 ppm. For the experiments, 5 ppm, 6.25 ppm, and 7.5 ppm final concentrations were used. For combined treatments, VD3 and AgNPs were mixed before use. In control samples only absolute ethanol was added in the same amount present in the experimental sample (50 µL/15 mL).

### 4.4. Cell Viability

MTT assay was used to test cellular viability, as previously reported [20]. HaCaT cells were seeded into 96-well plates at a density of 1 × 10^4^ cells/well with DMEM complete medium. After 24 h, culture medium was replaced with fresh, complete medium containing 100, 133, and 166 nM cholecalciferol or calcitriol and 5 ppm, 6.25 ppm, and 7.5 ppm AgNPs, or their combination, and the cells were incubated for 24. Then, MTT reagent was dissolved in PBS 1× and added to the culture at 0.5 mg/mL final concentration. After 3 h incubation at 37 °C, the supernatant was carefully removed, and formazan salt crystals were dissolved in 200 µL DMSO that was added to each well. The absorbance (OD) values were measured spectrophotometrically at 540 nm using an automatic microplate reader (Eliza MAT 2000, DRG Instruments, GmbH, Marburg, Germany). Each experiment was performed two times in triplicate. Cell viability was expressed as a percentage relative to the control cells.

### 4.5. Trypan Blue Exclusion Assay

The trypan blue exclusion assay was performed using a Countess™ (Invitrogen Srl, Milan, Italy) automated cell counter to test the live and death cell numbers, as previously reported [8]. Briefly, 50 μL of HaCaT cell suspension (5 × 10^4^/500 μL) was mixed with equal volumes of 0.4% trypan blue and loaded onto a Countess cell-counting chamber slide. Untreated cells were used as control. Images were captured by a camera and then analyzed with image analysis software to automatically measure cell count and viability.

### 4.6. In Vitro Wound Healing Assay

The wound healing assay was performed as previously reported [8] with CytoSelect™ Wound Healing Assay Kit (Cell Biolabs Inc., San Diego, CA, USA). HaCaT cells were seeded at 2 × 10^5^ concentration into 24-well tissue culture plate containing proprietary treated inserts in the plate wells with their “wound field” aligned in the same direction and incubated for 24 h to allow the cells to adhere and reach 80% confluence. After removing the inserts from the wells, the medium was carefully aspired. Then, the wells were washed twice with serum-free medium to remove dead cells and debris. Finally, the cells were treated with 133 nM VD3, 5 ppm AgNPs or 133 nM VD3 + 5 ppm AgNPs. Untreated cells were used as control. To analyze cell migration, representative images focused on the center of the wound field were captured by using inverted microscopy EUROMEX FE 2935 (ED Amhem, The Netherlands) equipped with a CMEX 5000 camera system (10× magnification). At least three fields for each condition were taken. Image acquisition and measurement of the distance between the two wound edges were performed by using ImageFocus software. The percentage of closure was calculated after 6 h, 12 h, and 18 h from treatment, by considering the initial cut, at 0 h, of 0.9 mm. Three sets of experiments in duplicates were performed.

### 4.7. Immunocytochemistry

Cells were cultured for 24 h in the absence or presence of 133 nM VD3 + 5 ppm AgNPs and then were centrifuged at 1000× *g* for 10 min. The pellets were fixed in 10% neutral phosphate-buffered formaldehyde solution for 24 h. Then, the cytoinclusion technique by the Cellient^®^ Automated Cell Block System (Hologic, Mississauga, ON, Canada) that rapidly creates a paraffin embedded cell block was used. Bond Dewax solution was used to remove paraffin from sections before rehydration and immunostaining on the Bond automated system (Leica Biosystems Newcastle, Ltd., Newcastle upon Tyne UK) as previously reported [16]. Immunostaining for cytokeratine (CK), vimentin (VIM), and α-smooth muscle actin (α-SMA) detection was performed by using specific antibodies and Bond Polymer Refine Detection—Leica Biosystems (Newcastle, Ltd., Newcastle upon Tyne, UK). The observations were performed by using inverted microscopy EUROMEX FE 2935 (ED Amhem, Netherlands) equipped with a CMEX 5000 camera system (40× magnification) [20].

### 4.8. Quantitative Real-Time RT-PCR

Cells cultured in the absence or presence of 133 nM VD3 + 5 ppm AgNPs were used for total RNA extraction performed using RNAqueous^®^-4PCR kit (Ambion Inc., Austin, TX, USA). RTqPCR was performed as previously reported [20] using Master Mix TaqMan Gene Expression and 7.300 RT-PCR instrument (Applied Biosystems), targeting genes in TaqManArray 96-well plate P/N: 4414250. VDR (Hs00172113_m). Cyclin D1 (CCND1, HS00765553), SM phosphodiesterase 4 (SMPD4, Hs04187047_g1), neutral ceramidase (ASAH2, Hs01015655_m1), and sphingosine kinase 2 (SPHK2, Hs01016543_g1) genes were tested. Glyceraldehyde-3-phosphate dehydrogenase (GAPDH, Hs99999905_m1) and 18S rRNA (S18, Hs99999901_s1) were used as housekeeping genes. mRNA relative expression levels were calculated as 2^−ΔΔ^*^C^*^t^ by comparing the results of VD3-treated sample with those of untreated samples.

### 4.9. Western Blotting

Cells were cultured for 24 h in the absence or presence of 133 nM VD3 + 5 ppm AgNPs for protein detection. About 40 µg of cell proteins were used for SDS-PAGE elec-trophoresis in 10% polyacrylamide lab gel. Proteins were transferred into nitrocellulose for 90 min as previously described [20]. The membranes were blocked for 30 min with 5% no-fat dry milk in PBS (pH 7.4) and incubated overnight at 4 °C with VDR or Cyclin D or nSMase or nCerase or SPHK2 specific antibodies. Anti-βtubulin antibody was used to normalize the data. The blots were treated with HRP-conjugated secondary anti-bodies for 90 min. Band detection was performed using an enhanced chemilumiescence kit from Amersham Pharmacia Biotech (Rainham, Essex, UK). A densitometric analysis was performed by Chemidoc Imagequant LAS500–Ge Healthcare-Life Science (Mi-lano, Italy).

### 4.10. Ultrafast Liquid Chromatography–Tandem Mass Spectrometry

Lipid extraction and analysis was performed as previously reported [39,40]. The pellets of the cells were suspended in Tris 10 mM, pH 7.4, and diluted with 1 mL methanol. Three milliliters of ultra-pure water and 3 mL methyl tert-butyl ether (MTBE) were added. Each sample was vortexed for 1 min and centrifuged at 3000× *g* for 5 min. The supernatant was recovered. The extraction with MTBE was repeated on the pellet and the supernatant was added to the first. The organic phase was dried under nitrogen flow and resuspended in 500 μL of methanol. The: 12:0 SM, 16:0 SM, 18:1 SM, 24:0 SM, 6:0 Cer, 8:0 Cer, 16:0 Cer, 18:0 Cer, 20:0 Cer, and 24:0 Cer standards were dissolved in chloroform/methanol (9:1 vol/vol) at 10 μg/mL final concentration. The stock solutions were stored at −20 °C. Working calibrators were prepared by diluting stock solutions with methanol to 500:0, 250:0, 100:0, and 50:0 ng/mL final concentrations. Twenty microliters of external standards or lipids extracted from serum with 12:0 SM and 6:0 Cer as internal standards (500 ng/mL) were injected after purification with specific nylon filters (0.2 μm). The analyses were carried out by using the Ultra Performance Liquid Chromatography system tandem mass spectrometer (Applied Biosystems, Italy). The lipid species were separated, identified, and analyzed as previously reported [21]. Liquid Chromatography system was Shimadzu Prominence UFLC, the pump was Shimadzu LC-20 AD, the detector was API 3200 linear triple quadrupole MS/MS, the injection valve was Valco valve, the autosampler was Schimadzu SR-20 AC HT, and the column temperature stabilizer was Schimadzu CTO-20A [21]. The samples were separated on a Phenomenex Kinetex phenyl-hexyl 100 A column (50 × 4.60-mm diameter, 2.6-μm particle diameter) with a precolumn security guard Phenomenex ULTRA phenyl-hexyl 4.6. Column temperature was set at 50 °C and the flow rate at 0.9 mL/min. Solvent A was 1% formic acid and solvent B was 100% isopropanol containing 0.1% formic acid. The run was performed for 3 min in 50% solvent B and then in a gradient to reach 100% solvent B in 5 min. The system needed to be reconditioned for 5 min with 50% solvent B before the next injection. The lipid species were identified by using positive turbo-ion spray (ESI) and modality multipole-reaction monitoring. Ion spray voltage was 5.4 kV, gas 1 was air, gas 2 was nitrogen, temperature was 650 °C, and the flow rate curtain gas was 40.5 L/h. The flow of the collision gas was maintained at 5.0 L/h. Data were acquired and processed using AnalystTM and Analyst 1.5 software in a Dell Precision T3400 personal computer with a Samsung ML-2851 MD graphical printer [21].

### 4.11. Statistical Analysis

Three experiments in duplicate were performed for each analysis. The data are expressed as mean ± S.D. Statistical differences were investigated by either unpaired t test or one-way ANOVA coupled with a Bonferroni post hoc test, in the case of more than two experimental groups (cell viability) and were set as * *p* < 0.05.

## Figures and Tables

**Figure 1 ijms-23-01410-f001:**
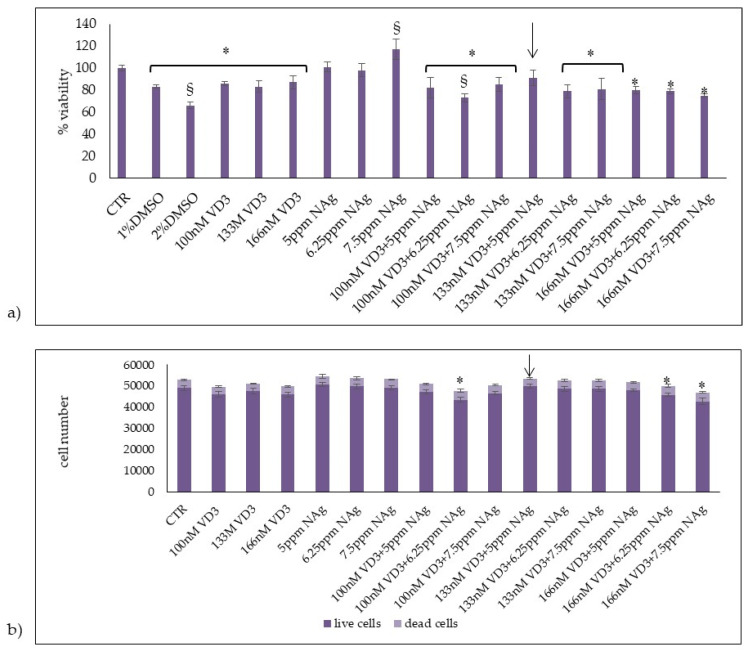
Effect of VD3, AgNPs (NAg), and their combination in HaCaT cells. Cells were cultured and treated as reported in Section 4. (**a**) Cell viability was measured by MTT assay. Values are reported as percentage viability of the control sample; 1% DMSO and 2% DMSO were used as positive controls. (**b**) Live and dead cell number were measured by trypan blue exclusion assay. Data are expressed as mean ± SD of three independent experiments performed in duplicate. Significance, * *p* < 0.05 versus the control sample analyzed by t test and ^§^
*p* < 0.05 analyzed by ANOVA coupled with Bonferroni post hoc test.

**Figure 2 ijms-23-01410-f002:**
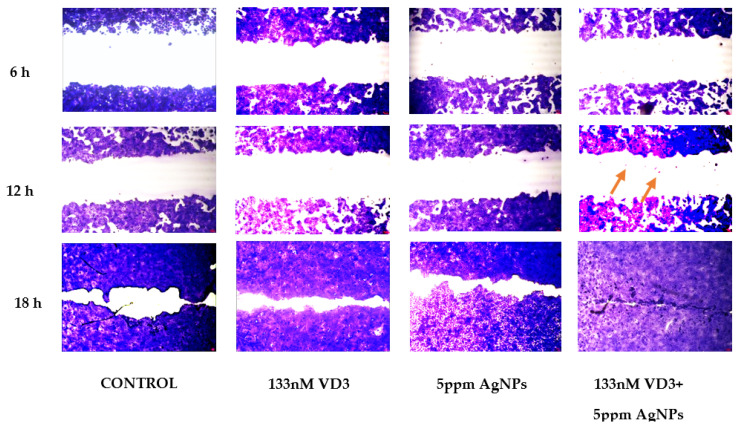
Effect of VD3, AgNPs, and their combination in wound closure. Wound healing assay was performed as reported in Section 4 and microscopy images were analyzed after 6, 12, and 18 h, compared to 0 h.

**Figure 3 ijms-23-01410-f003:**
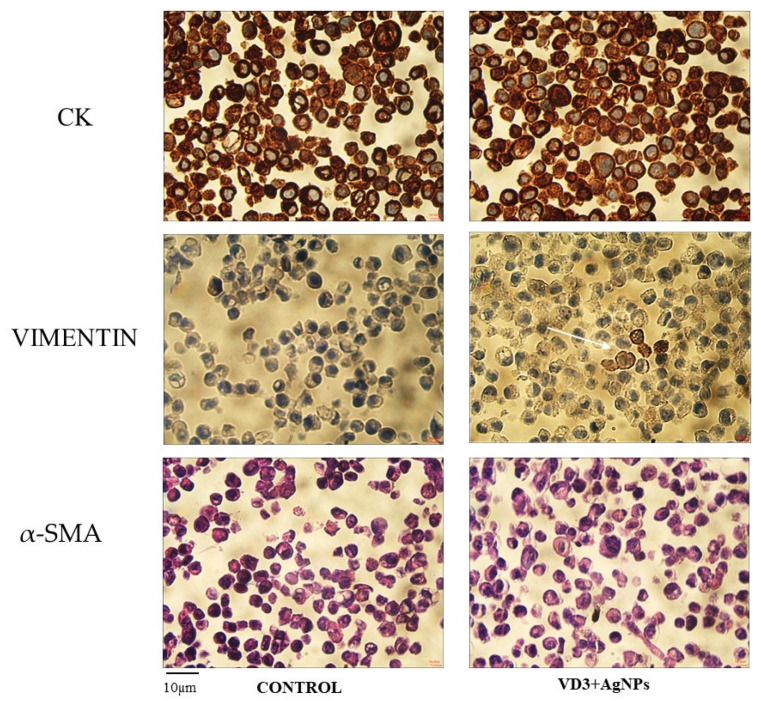
Immunocytochemical analysis of Cytokeratin (CK), Vimentin, and α-smooth muscle actin (α -SMA). Cells were cultured and treated as reported in Section 4. 40× magnification.

**Figure 4 ijms-23-01410-f004:**
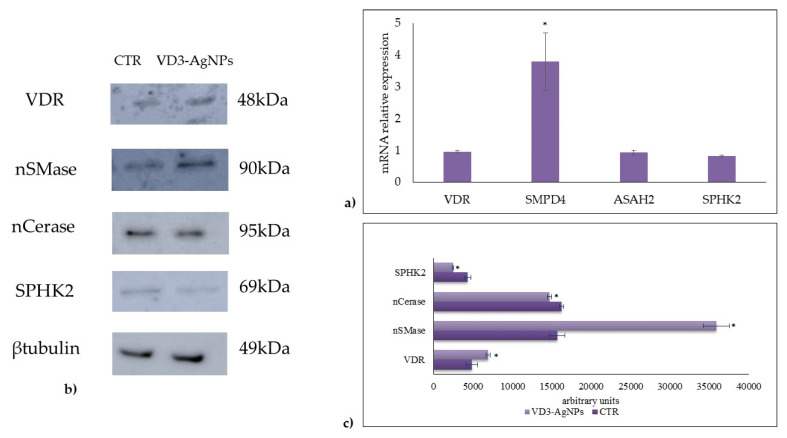
Effect of VD3 + AgNPs on gene and protein expression in HaCaT cells. Cells were cultured and treated as reported in Section 4. (**a**) Gene expression evaluated by RT-PCR; glyceraldehyde 3-phosphate dehydrogenase and 18S rRNA were used as housekeeping genes. (**b**) Western blotting; the position of proteins was indicated in relation to the position of molecular size standards, β-tubulin was used as loading control. (**c**) Area density evaluated by Chemidoc Imagequant TL software. VDR, vitamin D3 receptor; nSMase neutral sphingomyelinase; nCerase, neutral ceraminidase; and SPHK2, sphingosine kinase 2. Values of proteins, normalized with β-tubulin, are expressed as arbitrary units in control and experimental sample (treated with 133 nM VD3 + 5 ppm AgNPs) and represent the mean ± SD of three independent experiments performed in duplicate. Significance, * *p* < 0.05 versus the control sample.

**Figure 5 ijms-23-01410-f005:**
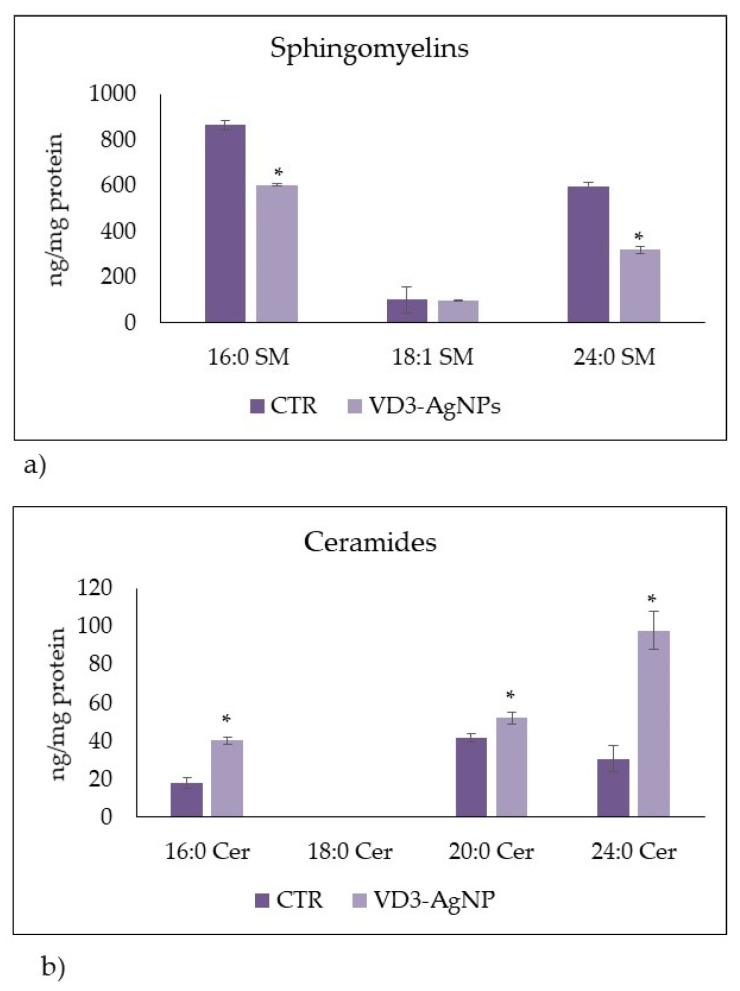
LC MS/MS analysis of SM and Cer species upon VD3-AgNP treatment. (**a**,**b**) Control (Ctr) and experimental (Ex) cells. The analysis was performed as reported in the Section 4. Data are expressed as ng/mg protein and represent the mean ± SD of two independent experiments performed in duplicate. * *p* < 0.05 versus control sample.

**Figure 6 ijms-23-01410-f006:**
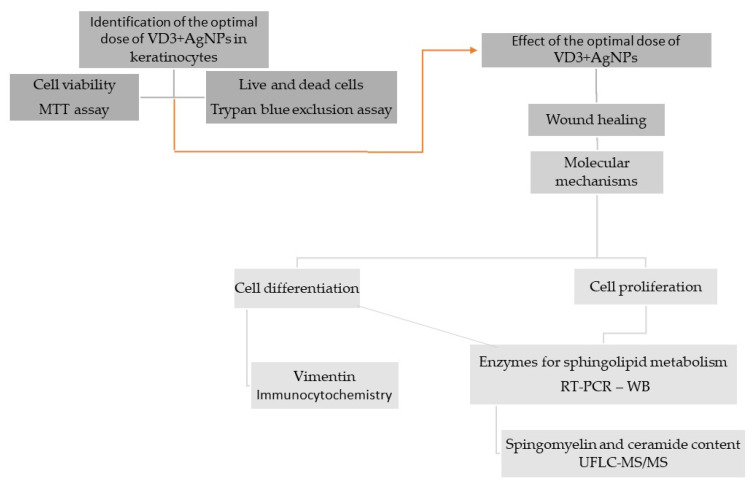
The experimental plan first envisaged the identification of the optimal dose of the VD3-AgNPs combination that did not induce cytotoxicity. Once the optimal dose was chosen, this was used to test its effectiveness on wound repair in vitro. Then, it was planned to study the molecular mechanisms on the differentiation and proliferation of keratinocytes, paying particular attention to the metabolism of sphingolipids.

## Data Availability

Not applicable.

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
