# Peer review of "The Effect of Vitamin D3 and Silver Nanoparticles on HaCaT Cell Viability"

_ijms, 2022, doi:10.3390/ijms23031410_

Round 1

Reviewer 1 Report

In the present study authors have described a combination of vitamin D3 plus (133nM) and silver nanoparticles (AgNPs 550nM, 30-50nm) that caused a complete wound healing in18hr in cell culture experiments. Authors previously shown that vitamin D3 has increased the wound healing but not resulting in a complete healing. The metabolism involved in the response to treatment, was studied based on the expression of gene encoding and in protein level.

The study is well planned and most of the conclusions are supported by the data. It is of high quality to be published in the International Journal of Molecular Sciences.

However, the following minor points are recommended to be corrected.

  1. Page 3, “As shown in figure 2, an open gap of 0.9 mm was created by a “wound” according to the manufacture’s instruction, and the reaction of the cells to fill the damaged area was inspected microscopically over time (6, 12, and 18 h).” a brief information and background about the manufacture instruction and how it theoretically applies should be added in the text.
  2. In the Figure 2. “Effect of VD3, AgNPs and their combination in wound closure, Wound healing assay was performed as reported in Material and Methods and microscopy images were analyzed after 6, 12, and 18 h respect to 0 h”.

This figure is missing the correlation of images corresponding to 18 hr, 0 hr and control sample.

  1. Page 9, “4.6. In Vitro Wound Healing Assay: To analyze cell migration, representative images focused on the center of the wound field were photographed according to manufacturer’s instructions. At least 3 fields for each condition were taken, and the numbers of migrating cells into scratched fields and the percentage of closure were calculated after 6h, 12h and 18h from treatment.” A clear description about the quantification of images that is based on cells migration should be added.
  2. Page 9 “highly stable aqueous colloidal solution of AgNPs of 30-50nm of diameter (5ppm, 6.25ppm and 7.5ppm).” Synthesis and preparation of AgNPs, characterisation of size range and quantification of AgNPs to ppm scale should be added.

Author Response

In the present study authors have described a combination of vitamin D3 plus (133nM) and silver nanoparticles (AgNPs 550nM, 30-50nm) that caused a complete wound healing in18hr in cell culture experiments. Authors previously shown that vitamin D3 has increased the wound healing but not resulting in a complete healing. The metabolism involved in the response to treatment, was studied based on the expression of gene encoding and in protein level.

The study is well planned and most of the conclusions are supported by the data. It is of high quality to be published in the International Journal of Molecular Sciences.

However, the following minor points are recommended to be corrected.

  1. Page 3, “As shown in figure 2, an open gap of 0.9 mm was created by a “wound” according to the manufacture’s instruction, and the reaction of the cells to fill the damaged area was inspected microscopically over time (6, 12, and 18 h).” a brief information and background about the manufacture instruction and how it theoretically applies should be added in the text.

Thank you for this observation. I’m sorry for this! It has already described in the “Materials and Methods” (lines295-304). Therefore, here I have corrected the sentence (line 98)

  1. In the Figure 2. “Effect of VD3, AgNPs and their combination in wound closure, Wound healing assay was performed as reported in Material and Methods and microscopy images were analyzed after 6, 12, and 18 h respect to 0 h”.

This figure is missing the correlation of images corresponding to 18 hr, 0 hr and control sample. In the control sample, the open gap remained unchanged after 6 h

the sentence was confused and was fixed. At time 0 the 0.9mm cut was made in all samples and the analyzes were carried out at 6h, 12h and 24h, as shown in figure 2.  In the control sample at 6h the cut remained  0.9mm (lines 96-100).

  1. Page 9, “4.6. In Vitro Wound Healing Assay: To analyze cell migration, representative images focused on the center of the wound field were photographed according to manufacturer’s instructions. At least 3 fields for each condition were taken, and the numbers of migrating cells into scratched fields and the percentage of closure were calculated after 6h, 12h and 18h from treatment.” A clear description about the quantification of images that is based on cells migration should be added.

The sentence has been clarified (lines 311-319).

  1. Page 9 “highly stable aqueous colloidal solution of AgNPs of 30-50nm of diameter (5ppm, 6.25ppm and 7.5ppm).” Synthesis and preparation of AgNPs, characterisation of size range and quantification of AgNPs to ppm scale should be added.

I'm sorry but this is owned by the supplier company (NanoBMat Company (Düsseldorf, Germany, line 268), we have to consider what is reported on the label. The initial concentration was reported (lines 287-280)

Reviewer 2 Report

The manuscript by Cataldi et al. reports on the effect of the combination of Silver Nanoparticles and Vitamin D3 on HaCaT cell viability. The topic of the paper is of interest for scientists working in different research fields and well fits with the scope of IJMS. This reviewer encourages authors to address the followings comments and revise the paper before further processing.

Below the comments as per article section. Some comments are reported as questions just for a faster and prompt reply by the authors.

Abstract

In its present form it is too literal. Authors should revise the section by inserting some key data of their study

Introduction

The section can be considered organized in three parts: Vitamin D3 (lines 32-56), Silver Nanoparticles (lines 57-61 and scope of the paper (lines 62-64). Although this reviewer thinks an added value short introduction section, a better connection between the three parts could help readers. Authors should consider briefly extend the AgNPs part and better focus the novelty of their study.

Results

Please revise English form in the first part of the section, especially in the first part (lines 67-73), since too spot sentences are reported here.

Discussion

The same as above, please revise English to be more fluent in reading.

More importantly, can authors compare their results with different combination of VD3 and nanomaterials to improve the interest of readers?

At lines 221-22 authors stated “It is possible to suppose a connection between nSMase and cell proliferation, previously demonstrated by Devillard et al”. This reviewer is wondering if this is the most appropriate lines to plate the sentence, and can authors extend this discussion for better connection with the remaining text?

Conclusions

Please revise the section. The main outcome of the study should be clear here.

Materials and Methods

No specific comments in this section.

References

Please check reference style according to the journal guidelines.

Author Response

The manuscript by Cataldi et al. reports on the effect of the combination of Silver Nanoparticles and Vitamin D3 on HaCaT cell viability. The topic of the paper is of interest for scientists working in different research fields and well fits with the scope of IJMS. This reviewer encourages authors to address the followings comments and revise the paper before further processing.

Below the comments as per article section. Some comments are reported as questions just for a faster and prompt reply by the authors.

Abstract

In its present form it is too literal. Authors should revise the section by inserting some key data of their study

The abstract was revised

Introduction

The section can be considered organized in three parts: Vitamin D3 (lines 32-56), Silver Nanoparticles (lines 57-61 and scope of the paper (lines 62-64). Although this reviewer thinks an added value short introduction section, a better connection between the three parts could help readers. Authors should consider briefly extend the AgNPs part and better focus the novelty of their study.

The introduction has been revised according to your suggestions

Results

Please revise English form in the first part of the section, especially in the first part (lines 67-73), since too spot sentences are reported here.

The English form in the first part of the “results” section has been revised

Discussion

The same as above, please revise English to be more fluent in reading.

More importantly, can authors compare their results with different combination of VD3 and nanomaterials to improve the interest of readers?

At lines 221-22 authors stated “It is possible to suppose a connection between nSMase and cell proliferation, previously demonstrated by Devillard et al”. This reviewer is wondering if this is the most appropriate lines to plate the sentence, and can authors extend this discussion for better connection with the remaining text?

“Discussion” section has been revised according to your suggestions

Conclusions

Please revise the section. The main outcome of the study should be clear here.

Conclusions have been revised

Materials and Methods

No specific comments in this section.

References

Please check reference style according to the journal guidelines.

References have been checked

Round 2

Reviewer 2 Report

Authors addressed all the previous comments and improved the paper, which is recommended for publication in its current form.

This manuscript is a resubmission of an earlier submission. The following is a list of the peer review reports and author responses from that submission.

Round 1

Reviewer 1 Report

This is an interesting article showing the wound healing activity of vitamin D3 plus silver nanoparticles. The authors provide data strongly suggesting a role of sphingolipids in this effect. In my opinion, the results are worth publishing, but the article would be much stronger after addressing the following comments:

Lines 132-143. The authors hypothesized that ceramide (Cer) derived from SM by nSMase action was catabolized by ceramidase (Cerase) to form sphingosine. This could be metabolized by sphingosine kinase 2 (SPHK2) to form sphingosine 1-phosphate (S1P), which would be involved in macrophagy motility during wound healing. In agreement, they found that VD3+AgNPs treatment induced an overexpression of SMPD4. However, nCerase and SPHK2 were not augmented. Why did not the authors examine the expression of other ceramidases and of SphK1? Regarding ceramidases, acid ceramidase (AC) and alkaline ceramidase 1 (ACER1) are the main isoforms in keratinocytes (J Lipid Res. 2006 May;47(5):1063-70. doi: 10.1194/jlr.M600001-JLR200. Epub 2006 Feb 13). I suggest determining whether AC, ACER1 and SphK1 change upon VD3+AgNPs treatment. Why did not the authors analyze SM, ceramides, sphingosine and S1P by LC/MS analysis?. I suggest these analyses to be performed to obtain data in support of their hypothesis. Regarding ceramides role in regulating VD3+AgNPs-induced wound healing, given the different roles of the distinct N-acylated ceramides at inducing proliferation or death, LC/MS will show if ceramides increase as a whole or in any specific species.

Line 168. “leading to the Cer degradation we determined that both nCerase and SPHK2 enzymes were downregulated”. The reduction in nCerase and Sphk2 is not statistically significant and thus this statement is not correct.

174-180. The authors hypothesize: “Mechanistically, it is possible to suppose that Cer might be responsible for epithelial mesenchymal transition demonstrated by the expression of vimentin in keratinocytes treated with VD3-AgNPs”. The authors should discuss what is reported about the role of Cer in epithelial mesenchymal transition (for instance (but not exclusively), Edmond et al., Oncogene. 2015 Feb 19;34(8):996-1005).

Minor issues:

Line 48. “shown to have potent effects anti-inflammatory and anti-cancer in the skin” should read “shown to have potent anti-inflammatory and anti-cancer effects in the skin”

Line 67. Replace “increasing and combined doses” by “increasing combined doses”

Line 81. Replace “in term of” by “in terms of”

Legend Fig. 1. Significance, * p < 0.05 versus the control sample. Please specify the statistical test used in the legend to avoid having to go to the Methods Section.

Lines 112-113/114-115. Replace “experimental sample” by “VD3+AgNPs treated samples”.

Figure 3. label experimental as VD3+AgNPs

Line 126. Replace “cyclin D1 protein (Fig. 4b,c), indicating an increase” by “in agreement with the observed increase in cell proliferation”

There are a number of typo errors that must be corrected.

Author Response

This is an interesting article showing the wound healing activity of vitamin D3 plus silver nanoparticles. The authors provide data strongly suggesting a role of sphingolipids in this effect. In my opinion, the results are worth publishing, but the article would be much stronger after addressing the following comments:

Lines 132-143. The authors hypothesized that ceramide (Cer) derived from SM by nSMase action was catabolized by ceramidase (Cerase) to form sphingosine. This could be metabolized by sphingosine kinase 2 (SPHK2) to form sphingosine 1-phosphate (S1P), which would be involved in macrophagy motility during wound healing. In agreement, they found that VD3+AgNPs treatment induced an overexpression of SMPD4. However, nCerase and SPHK2 were not augmented. Why did not the authors examine the expression of other ceramidases and of SphK1? Regarding ceramidases, acid ceramidase (AC) and alkaline ceramidase 1 (ACER1) are the main isoforms in keratinocytes (J Lipid Res. 2006 May;47(5):1063-70. doi: 10.1194/jlr.M600001-JLR200. Epub 2006 Feb 13). I suggest determining whether AC, ACER1 and SphK1 change upon VD3+AgNPs treatment. Why did not the authors analyze SM, ceramides, sphingosine and S1P by LC/MS analysis?. I suggest these analyses to be performed to obtain data in support of their hypothesis. Regarding ceramides role in regulating VD3+AgNPs-induced wound healing, given the different roles of the distinct N-acylated ceramides at inducing proliferation or death, LC/MS will show if ceramides increase as a whole or in any specific species.

Thank you very much for your comment that allowed me to understand how our experimental choices were not clear. We have focused on the prevalence of proliferative phenomena over differentiation ones. Only few cell differentiate but wound repair completely. Therefore, nCerase was evaluated as a molecule involved in the keratinocyte proliferation, as also reported in the article that you have indicated and that we have included in the paper [18].  SPK2 is involved in the salvage pathway of Ceramide by increasing it [19, 30, 31]. The results and their interpretation have been clarified (lines 160-171, 230-255)

Thank you very much again for suggesting the lipidomic experiments that complemented our results (Lines 192-208, figure 5, 400-432). The new author who performed the analysis has been added.

Line 168. “leading to the Cer degradation we determined that both nCerase and SPHK2 enzymes were downregulated”. The reduction in nCerase and Sphk2 is not statistically significant and thus this statement is not correct.

Our statistical analysis show that differences are statically significant, see figure 4 b.

174-180. The authors hypothesize: “Mechanistically, it is possible to suppose that Cer might be responsible for epithelial mesenchymal transition demonstrated by the expression of vimentin in keratinocytes treated with VD3-AgNPs”. The authors should discuss what is reported about the role of Cer in epithelial mesenchymal transition (for instance (but not exclusively), Edmond et al., Oncogene. 2015 Feb 19;34(8):996-1005).

It has been included in the discussion (Lines 262,263)

Minor issues:

Line 48. “shown to have potent effects anti-inflammatory and anti-cancer in the skin” should read “shown to have potent anti-inflammatory and anti-cancer effects in the skin”

It has been corrected (line 54)

Line 67. Replace “increasing and combined doses” by “increasing combined doses”.

The statement has been clarified (lines 73, 74)

Line 81. Replace “in term of” by “in terms of”

It has been corrected (line 88)

Legend Fig. 1. Significance, * < 0.05 versus the control sample. Please specify the statistical test used in the legend to avoid having to go to the Methods Section.

It has been reported in the legend (lines 98,99)

Lines 112-113/114-115. Replace “experimental sample” by “VD3+AgNPs treated samples”.

It has been performed along the text and in all figures

Figure 3. label experimental as VD3+AgNPs

It has been made

Line 126. Replace “cyclin D1 protein (Fig. 4b,c), indicating an increase” by “in agreement with the observed increase in cell proliferation”

Cyclin D1 has been deleted from the paper

There are a number of typo errors that must be corrected.

The type errors have been revised along the text

Reviewer 2 Report

The present study evaluated the effect of vitamin D3 incorporated to silver nanoparticles in an in vitro system of wound healing. During my review process I found several concerns that should be addressed by the authors to improve the final version of the manuscript.

  1. A figure describing the study's design should be included.
  2. The statistical analysis is poor and the t-test is only indicated for evaluating some data from the study. Please, consult to an statistical adviser to improve the evaluation of your data.
  3. The discussion section is poor and the results are vaguely discussed.
  4. Please, add a conclusion section.

Author Response

The present study evaluated the effect of vitamin D3 incorporated to silver nanoparticles in an in vitro system of wound healing. During my review process I found several concerns that should be addressed by the authors to improve the final version of the manuscript.

  1. A figure describing the study's design should be included.

The experimental design has been included at the beginning of Materials and Methods(lines 274-284)

  1. The statistical analysis is poor and the t-test is only indicated for evaluating some data from the study. Please, consult to an statistical adviser to improve the evaluation of your data.

The statistical analysis has been revised by an expert and data have been reported (legend figure 1 lines 98, 99 and lines 434-437)

  1. The discussion section is poor and the results are vaguely discussed.

Discussion has been rewritten (lines 220-265)

  1. Please, add a conclusion section.

Conclusion section has been added (lines 268-272)

Reviewer 3 Report

In the manuscript „Vitamin D3 coupled silver nanoparticles for effective wound healing“, the authors have explored very simple model to show that that „a combination of 133nM vitamin D3 plus 550 nM silver nanoparticles of 30-20 nm of diameter induced a complete wound healing only after 18h“.

This is a superficially written material, with the data obtained on only one cell line; HaCaT - the immortalized human keratinocytes.

The authors did not describe, and they should have, the experimental procedures in more details. Indeed, very important data are missing.

For example:

  1. The authors did not state the solvent for the silver particles and vitamin D. Whatever the solvent was (DMSO?) it needed to be included in the wound healing experiment, WBs and RT-qPCR. Without having a “vehicle control” one cannot make any conclusion;
  2. Was the anti-CK antibody used specific, or it was just a pan-CK- antibody? That should have been stated. No single specific information on antibodies and procedures performed;
  3. When describing the ICC – immunocytochemistry, the authors claim that „Bond Dewax solution was used to remove paraffin from sections before rehydration and immunostaining on the Bond automated system“. Which paraffin in ICC? Not possible.
  4. It is also interesting that the authors claim working with TaqMan SNP Genotyping Assay (section 4.1). That was clearly not the case. They worked with the different type of TaqMan.
  5. WBs: Figure 4. “… the position of proteins was indicated in relation to the position of molecular size standards”. No protein standards presented. They should be presented.

Claiming that the healing properties of tested substances exclusively work through increased proliferation which is consequential to CCND1 increase, cannot be claimed without performing: a) BrdU assay; b) CCND1 silencing.

I believe, that for IJMS, the authors need to extend their focus to the mechanicistic aspects of phenomena observed. Without it, this is just one observational study.

Thank you.

Author Response

In the manuscript „Vitamin D3 coupled silver nanoparticles for effective wound healing“, the authors have explored very simple model to show that that „a combination of 133nM vitamin D3 plus 550 nM silver nanoparticles of 30-20 nm of diameter induced a complete wound healing only after 18h“.

This is a superficially written material, with the data obtained on only one cell line; HaCaT - the immortalized human keratinocytes.

The authors did not describe, and they should have, the experimental procedures in more details. Indeed, very important data are missing.

Thank you very much for your observations that really improved the manuscript

All experimental part has been accurately revised

For example:

  1. The authors did not state the solvent for the silver particles and vitamin D. Whatever the solvent was (DMSO?) it needed to be included in the wound healing experiment, WBs and RT-qPCR. Without having a “vehicle control” one cannot make any conclusion;

Thank you very much for this observation. It has been included (lines 319-323)

  1. Was the anti-CK antibody used specific, or it was just a pan-CK- antibody? That should have been stated. No single specific information on antibodies and procedures performed;

It has been reported (line 298, 368)

  1. When describing the ICC – immunocytochemistry, the authors claim that „Bond Dewax solution was used to remove paraffin from sections before rehydration and immunostaining on the Bond automated system“. Which paraffin in ICC? Not possible.

We used a cytoinclusion technique by the Cellient® Automated Cell Block System (Hologic, Mississauga, Canada) that permits the inclusion of the cells in paraffin. The characteristics of the experiment has been reported (Lines 362-365)

  1. It is also interesting that the authors claim working with TaqMan SNP Genotyping Assay (section 4.1). That was clearly not the case. They worked with the different type of TaqMan.

It was an error and has been corrected (Lines 304,305)

  1. WBs: Figure 4. “… the position of proteins was indicated in relation to the position of molecular size standards”. No protein standards presented. They should be presented.

All original WB with protein standards were submitted as support materials during the first submission. I believe the pictures should be visible to you

Claiming that the healing properties of tested substances exclusively work through increased proliferation which is consequential to CCND1 increase, cannot be claimed without performing: a) BrdU assay; b) CCND1 silencing.

Experiments for CCND1 and cyclin D1 detection have been deleted. Proliferation has been evident by the complete wound repair, as previous paper [8]

I believe, that for IJMS, the authors need to extend their focus to the mechanicistic aspects o f phenomena observed. Without it, this is just one observational study.

All paper has been revised focusing the attention on sphingolipid metabolism as pathway involved in the response to VD3+AgNPs treatment. Experimental design has been included (lines 274-284).

Round 2

Reviewer 2 Report

All my concerns were well addressed.

Author Response

Thank you very much

Reviewer 3 Report

In the revised version of the manuscript „Vitamin D3 coupled silver nanoparticles for effective wound healing“, the authors did not offer convincing experimental proofs for their claims:

The authors claim a synergistic cellular effect, when combining 133nM VD3 + 5mpp AgNPs. I do not see it, neither on Fig 1a, neither on Fig. 1b. What I see is the effect which is less or equal to the effect obtained with 5 ppm AgNPs alone.

How can the authors be certain that combination of 133nM VD3 + 5mpp AgNPs induces increased transcription rate of SMPD4, without exploring the SMPD4 transcription after treatment with each compound, separately?

The possibility that only one of these compounds (VD3 or AgNPs) induces the effect measured at the level of mRNA/protein is not excluded at all. Since the authors refer to their previous work, I read some of the cited articles.

The authors claim: „It has been demonstrated that VD3 stimulated its own receptor (VDR) [16]. Thus, we analyzed VDR gene expression and protein level. The results showed that VD3 combined with AgNPs unchanged VDR gene expression (Fig.4a). Differently, the treatment induced an increase in VDR protein level (Fig.4 bc), indicating a possible slowdown of VDR break down due to its intracellular need to promote the response to VD3.“

The reference 16 relates to the response of embryonic hippocampal cells. Whether the same mechanism applies to immortalized keratinocytes remains to be seen, because VDR level can be increased by VD3 through transcriptional activation or VDR stabilization in a cell-line specific manner.

Indeed, in their previous work, the authors have shown that exposure of HaCaT to 100 mM and 200 mM VD3 during 24 hours increases VDR gene activity for about 1.5-1.6 and 4.5 fold change, respectively (the article cited as #8). Based on what is shown here, in this manuscript (Fig. 4A), should we conclude that AgNPs decreases activity of the VDR promoter? Or not?

The original WB showing VDR is of extremely low quality. The target band has an intensity equal to the two prominent background spots (attached).

The authors did not exclude stimulatory effect of ethanol on ceramide synthesis, if, as they claim, „control“ presents cells treated with ethanol. The „real“ control cells should be untreated cells, while „vechicle control“ are „ethanol treated“, if ethanol was used as a solvent for Vitamin D3.

Finally, the image showing the wound closure for 133nM VD3 after 12 hours is EXACTLY THE SAME as the image showing "control" after the same time period.

Thank you.

Author Response

In the revised version of the manuscript „Vitamin D3 coupled silver nanoparticles for effective wound healing“, the authors did not offer convincing experimental proofs for their claims:

The authors claim a synergistic cellular effect, when combining 133nM VD3 + 5mpp AgNPs. I do not see it, neither on Fig 1a, neither on Fig. 1b. What I see is the effect which is less or equal to the effect obtained with 5 ppm AgNPs alone.

We are very sorry that our experimentation is not convincing also because these observations were not evident in the first referee work.

Figure 1 was not intended to demonstrate synergism but to see if single doses of AgNPs or VD3 or combined doses could have a cytotoxic effect. We chose from the AgNPs + VD3 doses the one that induced a higher value of cell viability.

In figure 2 the synergistic effect is very evident since the complete closure of the wound is obtained only with the combined treatment

How can the authors be certain that combination of 133nM VD3 + 5mpp AgNPs induces increased transcription rate of SMPD4, without exploring the SMPD4 transcription after treatment with each compound, separately?

The objective was not to evaluate the combined effect with respect to the single components but the effect of the combined with respect to the control because the aim was to establish what the combined did at the molecular level since it was able to close the wound. As reported in the discussion, we already knew that treatment with VD3 alone was stimulating sphingomyelinase, we now wanted to see if the combined product had any effect on SM metabolism. We were no longer interested in the effect of the single treatment because it did not lead to complete closure

The possibility that only one of these compounds (VD3 or AgNPs) induces the effect measured at the level of mRNA/protein is not excluded at all. Since the authors refer to their previous work, I read some of the cited articles.

The authors claim: „It has been demonstrated that VD3 stimulated its own receptor (VDR) [16]. Thus, we analyzed VDR gene expression and protein level. The results showed that VD3 combined with AgNPs unchanged VDR gene expression (Fig.4a). Differently, the treatment induced an increase in VDR protein level (Fig.4 bc), indicating a possible slowdown of VDR break down due to its intracellular need to promote the response to VD3.“

The reference 16 relates to the response of embryonic hippocampal cells. Whether the same mechanism applies to immortalized keratinocytes remains to be seen, because VDR level can be increased by VD3 through transcriptional activation or VDR stabilization in a cell-line specific manner.

Indeed, in their previous work, the authors have shown that exposure of HaCaT to 100 mM and 200 mM VD3 during 24 hours increases VDR gene activity for about 1.5-1.6 and 4.5 fold change, respectively (the article cited as #8). Based on what is shown here, in this manuscript (Fig. 4A), should we conclude that AgNPs decreases activity of the VDR promoter? Or not?

Yes, this is highly probable, the combined effect does not reflect the effect of the single components and for this reason the single components have not been tested because the aim was to evaluate the effect of VD3 + AgNPs which induces a complete closure of the wound.

This happens very often when two combined substances are used. On the other hand, this is also evident in vitality. 133nM VD3 induces a cellular viability of 83+5.1, 5 ppm of 101+4.2 and 133nM + 5ppm a viability of 91+7.2. The differences are not significant but the trend is evident, the vitality value obtained from the combined treatment is intermediate compared to that obtained with the single treatments

The regulation of VDR by VD3 was studied by other authors in other cells:

The impact of the vitamin D-modulated epigenome on VDR target gene regulation.

Nurminen V, Neme A, Seuter S, Carlberg C.Biochim Biophys Acta Gene Regul Mech. 2018 Aug;1861(8):697-705. doi: 10.1016/j.bbagrm.2018.05.006

Activation of the vitamin D receptor transcription factor stimulates the growth of definitive erythroid progenitors.

Barminko J, Reinholt BM, Emmanuelli A, Lejeune AN, Baron MH.Blood Adv. 2018 Jun 12;2(11):1207-1219. doi: 10.1182/bloodadvances.2018017533

The original WB showing VDR is of extremely low quality. The target band has an intensity equal to the two prominent background spots (attached).

You can see below the image of another WB performed in the same experiment and reported as "original WB" and reported also in the attached file. Here, in the experimental sample the VDR seems much more expressed but, by evaluating the densitometries and referring the values ​​to those of the tubulin, it falls within the standard deviation. This is a WB considered in densitometric analysis. 

The authors did not exclude stimulatory effect of ethanol on ceramide synthesis, if, as they claim, „control“ presents cells treated with ethanol. The „real“ control cells should be untreated cells, while „vechicle control“ are „ethanol treated“, if ethanol was used as a solvent for Vitamin D3.

As ethanol is present in the experimental sample per vehicle, correct control are the cells treated only with the vehicle, in our opinion. In any case the quantity added is only 50µl/15mL. This was included in the article (lines 334, 335).

Finally, the image showing the wound closure for 133nM VD3 after 12 hours is EXACTLY THE SAME as the image showing "control" after the same time period.

Sorry, this is our mounting error, we uploaded the same image twice, The corrected image has been included. Really thank you very much for this comment

Round 3

Reviewer 3 Report

I have no further comments for authors. I thank them for their replies.